# Viromics Reveals the High Diversity of Viruses from Fishes of the Tibet Highland

Yuan Xi,[a] Xiaojie Jiang,[a] Xinrui Xie,[b] Min Zhao,[a] Han Zhang,[a] Kailin Qin,[a] Xiaochun Wang,[a] Yuwei Liu,[a] Shixing Yang,[a] ⬤Quan Shen,[a] Likai Ji,[a] Peng Shang,[b] ⬤Wen Zhang,[a] ⬤Tongling Shan[c]

[a]Department of Microbiology, School of Medicine, Jiangsu University, Zhenjiang, Jiangsu, China
[b]Animal Science College, Tibet Agriculture and Animal Husbandry University, Nyingchi, Tibet, China
[c]Shanghai Veterinary Research Institute, Chinese Academy of Agricultural Sciences, Shanghai, China

Yuan Xi, Xiaojie Jiang and Xinrui Xie contributed equally to this work. Author order was determined by the corresponding authors after negotiation.

**ABSTRACT** Aquaculture is important for food security and nutrition. The economy has recently been significantly threatened and the risk of zoonoses significantly increased by aquatic diseases, and the ongoing introduction of new aquatic pathogens, particularly viruses, continues to represent a hazard. Yet, our knowledge of the diversity and abundance of fish viruses is still limited. Here, we conducted a metagenomic survey of different species of healthy fishes caught in the Lhasa River, Tibet, China, and sampled intestinal contents, gills, and tissues. To be more precise, by identifying and analyzing viral genomes, we aim to determine the abundance, diversity, and evolutionary relationships of viruses in fish with other potential hosts. Our analysis identified 28 potentially novel viruses, 22 of which may be associated with vertebrates, across seven viral families. During our research, we found several new strains of viruses in fish, including papillomavirus, hepadnavirus, and hepevirus. Additionally, we discovered two viral families, *Circoviridae* and *Parvoviridae*, which were prevalent and closely related to viruses that infect mammals. These findings further expand our understanding of highland fish viruses and highlight the emerging view that fish harbor large, unknown viruses.

**IMPORTANCE** The economy and zoonoses have recently been significantly threatened by aquatic diseases. Yet, our knowledge of the diversity and abundance of fish viruses is still limited. We identified the wide genetic diversity of viruses that these fish were harboring. Since there are currently few studies on the virome of fish living in the Tibet highland, our research adds to the body of knowledge. This discovery lays the groundwork for future studies on the virome of fish species and other highland animals, preserving the ecological equilibrium on the plateau.

**KEYWORDS** viral metagenomics, fishes, phylogenetic analysis, Tibetan Plateau

**Ad Hoc Peer Reviewers** ⬤Kanchan Bhardwaj, Manav Rachana International Institute of Research and Studies; Santanu Chattopadhyay, Rajiv Gandhi Centre for Biotechnology

Address correspondence to Wen Zhang, z0216wen@yahoo.com, Peng Shang, nemoshpmh@126.com, or Tongling Shan, shantongling@shvri.ac.cn.

The authors declare no conflict of interest.

[This article was published on 23 May 2023 with errors in the supplemental material. The errors were corrected in the current version, posted on 30 May 2023.]

Aquaculture is important for food security and nutrition since it provides vital bioavailable micronutrients in addition to providing roughly 17% of animal protein consumed globally and even more than 50% in some Asian nations (1). Until now, the development of aquatic production has been hampered by various disease-causing viruses and bacteria (2). Statistically speaking, viruses constitute a larger proportion of emerging pathogens in wildlife than bacteria do, and cross-species transmission of RNA viruses is more prevalent than that of DNA viruses (3). Among vertebrates, there is a clear sampling bias toward mammals and birds (4), although fish show significant diversity and abundance, accounting for approximately half of the total number of vertebrate species described (5). Viruses are the most prevalent biological organism on Earth while being the most basic form of life (6). Viral metagenomics has been

liberated from the constraints of culture methods and genetic marker analysis since the adoption of next-generation sequencing, and the quantity of metagenomic samples reflecting the aquatic environment has increased dramatically (7). As a result of the development of viral metagenomics, new and more fish-infecting viruses, such as those belonging to the families *Hepeviridae*, *Circoviridae*, and *Papillomaviridae*, are appearing (8–10). Nevertheless, until recently, barely 16 fish papillomavirus genomes have been identified (10), demonstrating the potential diversity of fish papillomaviruses in uncollected samples. Meanwhile, fish—as lower vertebrates—are of interest for their viruses and their relevance to the evolution of arthropod viruses (5). Undoubtedly, because fish are loaded with numerous unidentified viruses, it is imperative to identify the virus genera that infect different fish species to provide a relevant evolutionary perspective for the viruses that infect vertebrates and invertebrates in addition to examples of host jumping (11).

The risk of environmental contamination and the emergence of fish and aquatic zoonotic illnesses in humans are rising as a result of the expanding worldwide population and possible global aquaculture and fish trade. It is worth noting that fish can host viruses that are quite similar to known human infections, including members of the *Arenaviridae*, *Filoviridae*, and *Hantaviridae* families. In humans, these viruses are responsible for severe illnesses such as Lassa hemorrhagic fever, Ebola virus disease, and Hantavirus pulmonary syndrome that can cause significant harm to patients (4). This suggests that these previously mammalian-dominated pathogenic viruses exist in aquatic vertebrates as closely related viruses. Influenza-like viruses have also been discovered in fish, with the virus in skates being the one that is most similar to the human influenza B virus (4). Also, viruses of genera previously associated with vector-borne viruses were found in fish, indicating the association of lower vertebrates with invertebrates. Fish viruses spread primarily horizontally through excrement, polluted water, or the consumption of raw wild fish products (12). Consumption of raw fish has increased over the past few years, and with that, there has been a steady stream of human illnesses that have been linked to it, including acute gastroenteritis brought on by noroviruses (NoVs) (12, 13). Additionally, eating raw fish that has been contaminated with hepatitis A and E viruses can result in hepatitis and, in rare instances, liver failure or even death (14).

Fish viral communities have also been the subject of numerous recent research. The loss of less prevalent or short-lived viruses may occur as a result of two factors: first, sick or market fish were purchased rather than river fish being caught (15, 16); second, there were few samples of other tissues, with the majority of samples being gills and liver (17). In addition, the majority of the collecting sites were in non-highland bays or ocean areas, and many other areas, such as highland rivers, are yet unknown.

The Yarlung Tsangpo River (YTR), which in turn provides water sources for about 1.4 billion people in Asia, is one of the seven greatest Asian rivers, and the Tibetan Plateau is known as the "Water Tower of Asia" because of this (18, 19). The Lhasa River (LR), the largest tributary of the YTR, is at a high altitude and flows through Lhasa, the most populous city in Tibet, China. LR, as a drinking water source in the basin, has a relatively high population and economic level, and in recent years has become more vulnerable to human activities (20, 21). The LR is also a migratory wintering ground for many birds, including black-necked cranes, spotted geese, and red ducks, which provides the possibility of virus transmission.

To address important topics in fish viral ecology and evolution, we conducted a metagenomic survey of healthy fishes from LR and sampled intestinal contents, gills, and tissues (liver, muscle, swim bladder, spleen, heart, and brain). In particular, the composition and distinct alterations of viral ecological communities among the gills, intestinal contents, and tissues of virus-infected fish were investigated using viral metagenomic analysis. The genetic diversity of the predominant virus groups, including fish novel viruses, was investigated utilizing the phylogenetic analysis to ascertain the link between newly identified fish viruses and viruses that already infect vertebrates or invertebrates.

## RESULTS

**Overview of fish virome.** Metagenomic sequencing was performed on 18 pools that were manually segmented from 295 samples taken from fish in LR, resulting in a total of 20,156,456 raw reads with an average percentage of GC content (GC%) of 45.3%. Assemblies from the metagenomes were screened and compared to the GenBank nonredundant (nr) viral protein families-based database with blastx (E value of $<10^{-5}$) and identified 197,306 viral reads (0.979%). The species richness of all the fish samples was revealed by the species rarefaction and accumulation curves. In most of the 18 pools, the observed virus species flatten out, so it can be assumed that the sequencing depth has covered all species in the sample and increasing the sequencing data can no longer increase diversity (Fig. 1A). Despite the wide variation in viral taxonomy among all samples, the cumulative curve of viral species was saturated, demonstrating that viral communities in the fish were reasonably sufficiently sampled (Fig. 1B). The accumulation curves also demonstrated that there are likely to be over 400 different virus species spread throughout 18 pools.

Taxonomic analyses determined a total of 28 specific and nearly complete viral genomic sequences obtained by *de novo* assembly and reference mapping, including 19 DNA viruses and 3 RNA viruses that were known to infect vertebrates––*Circoviridae* ($n = 2$), CRESS DNA (i.e., circular Rep-encoding single-stranded DNA [ssDNA] viruses) ($n = 7$), *Hepadnaviridae* ($n = 1$), *Parvoviridae* ($n = 8$), *Papillomaviridae* ($n = 1$), *Hepeviridae* ($n = 1$), *Caliciviridae* ($n = 1$), and *Picornaviridae* ($n = 1$)––while the remaining 6 genomes were assigned as those whose members were not closely related to those known to infect vertebrates, including *Iflaviridae* ($n = 1$) and "unclassified picorna-like virus" ($n = 5$).

**Viral diversity and comparison in the viral communities.** A heat map was constructed to highlight the spread of the virus among different isolates of fish and contrast dynamic changes. Viral reads for which no further analysis has been performed were divided into 27 viral families at the family level, comprising 11 double-stranded DNA (dsDNA) viral families, 5 single-stranded DNA (ssDNA) viral families, 1 dsRNA viral family, and 10 ssRNA viral families (Fig. 2A). The *Circoviridae* family had a significantly higher number of viral reads than any of the other viral families (66,818 reads, accounted for 33.87%), making up a significant percentage of the virome. The legend to Fig. 2A states that the "intestinal contents" group has the greatest viral abundance and the "gill" group the least. For read abundance of different virus families in each pool of intestinal contents, those belonging to the *Circoviridae* and *Parvoviridae* families had the highest proportion of reads (Fig. 2B). Furthermore, the same is true for each pool of gills, except for the pools Fish093 and Fish098. However, it is notable that the *Retroviridae* family of viruses had the largest amount of reads in each pool of tissues. In the three grouped viral communities, a total of 617 virus species were found at the species level. The viral community obtained from fish tissues included the second-highest number of viral species after the viral community derived from fish intestinal contents. Meanwhile, 211 and 88 unique species, respectively, were present in the two viral communities in significant numbers (Fig. 2C). Only 7% to 18% of each species' total numbers were represented by the 23 virus species found in the three viral communities, indicating that each viral community has a substantial number of individual virus species that exhibit its own unique viral community characteristics. For additional investigation, the bubble diagram depicts the top 5 most prevalent virus species in each of the 3 virus communities: the viral community in the intestinal contents has the highest abundance of these virus species, which are primarily members of the *Circoviridae* and *Parvoviridae* families. In addition, among the viral communities derived from intestinal contents, species of *Lepidopteran iteradensovirus 5* of the *Parvoviridae* family and the *Circovirus-like* genome DHCV-6 of the *Circoviridae* family were particularly abundant, suggesting that these viral species may possess unique viral community structure. More precisely, we ranked the number of species in the 18 pools and presented them in a Nightingale rose diagram, which shows the highest abundance of species in pool Fish100 and the lowest abundance of species in pool Fish094 (Fig. 2C; see Table S1 in the supplemental material).

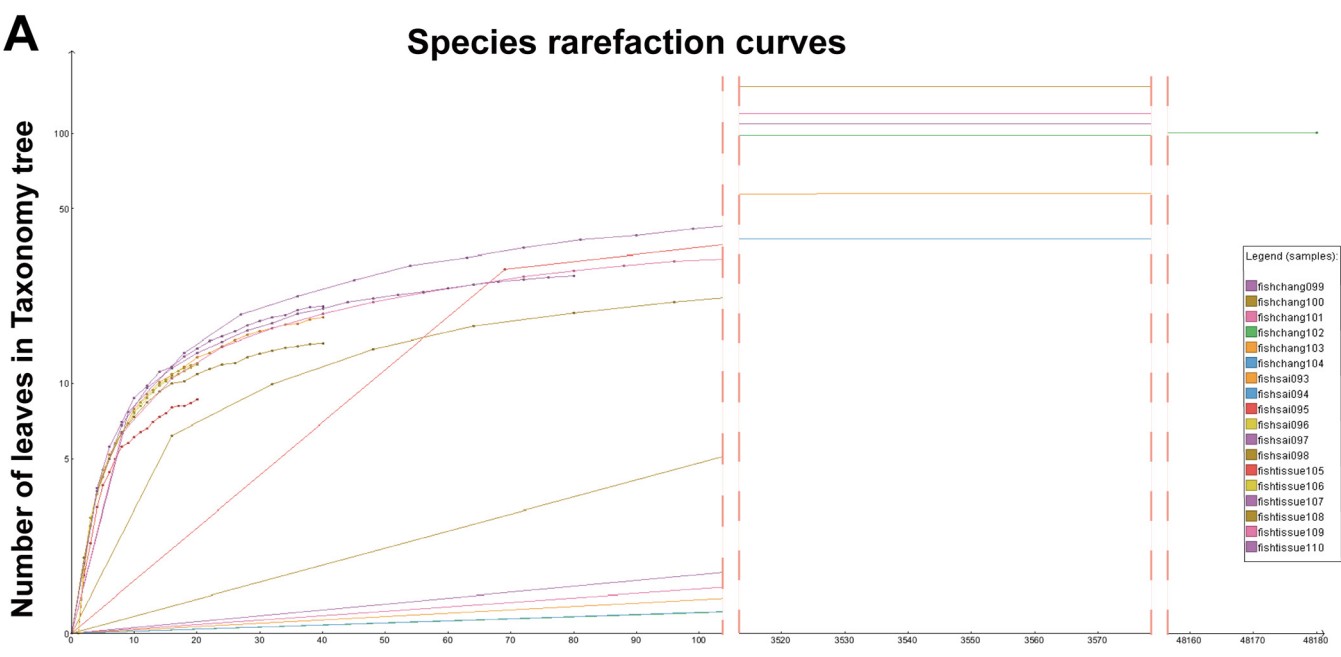

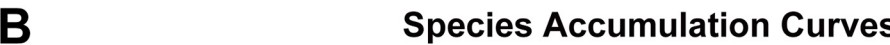

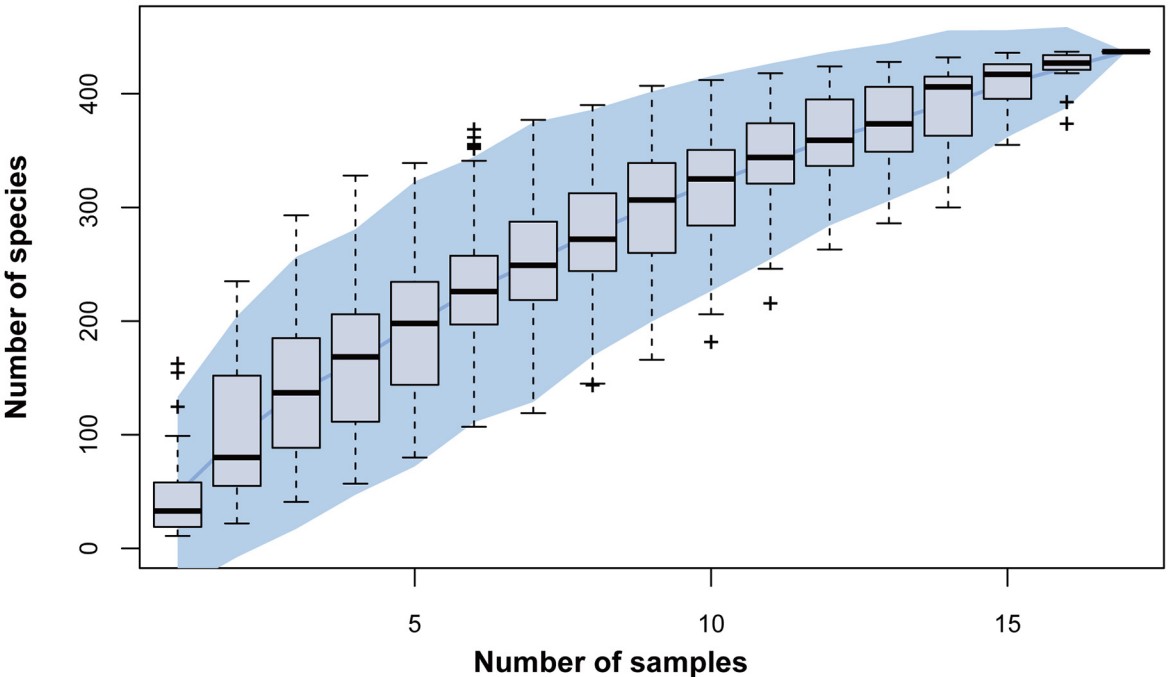

**FIG 1** Diversity of viral species in the 18 pools. (A) Species rarefaction curve annotated by MEGAN v.6.22.2 after log-scale transformation. Legends are displayed to the side of the panel. (B) Accumulation curve of viral species in the fish metagenomes. Error bars represent the range, and the blue area in the background represents the 95% confidence interval.

In order to explore the comparison of viral communities, the Shannon index is used to reflect $\alpha$ diversity and further observe the differences in the viral composition of intestinal contents, gills, and tissues. The viral communities between the intestinal contents and the other two groups differed significantly ($P < 0.05$ or $P < 0.01$), although there was no significant variation in the viral composition between the gills and tissues, as

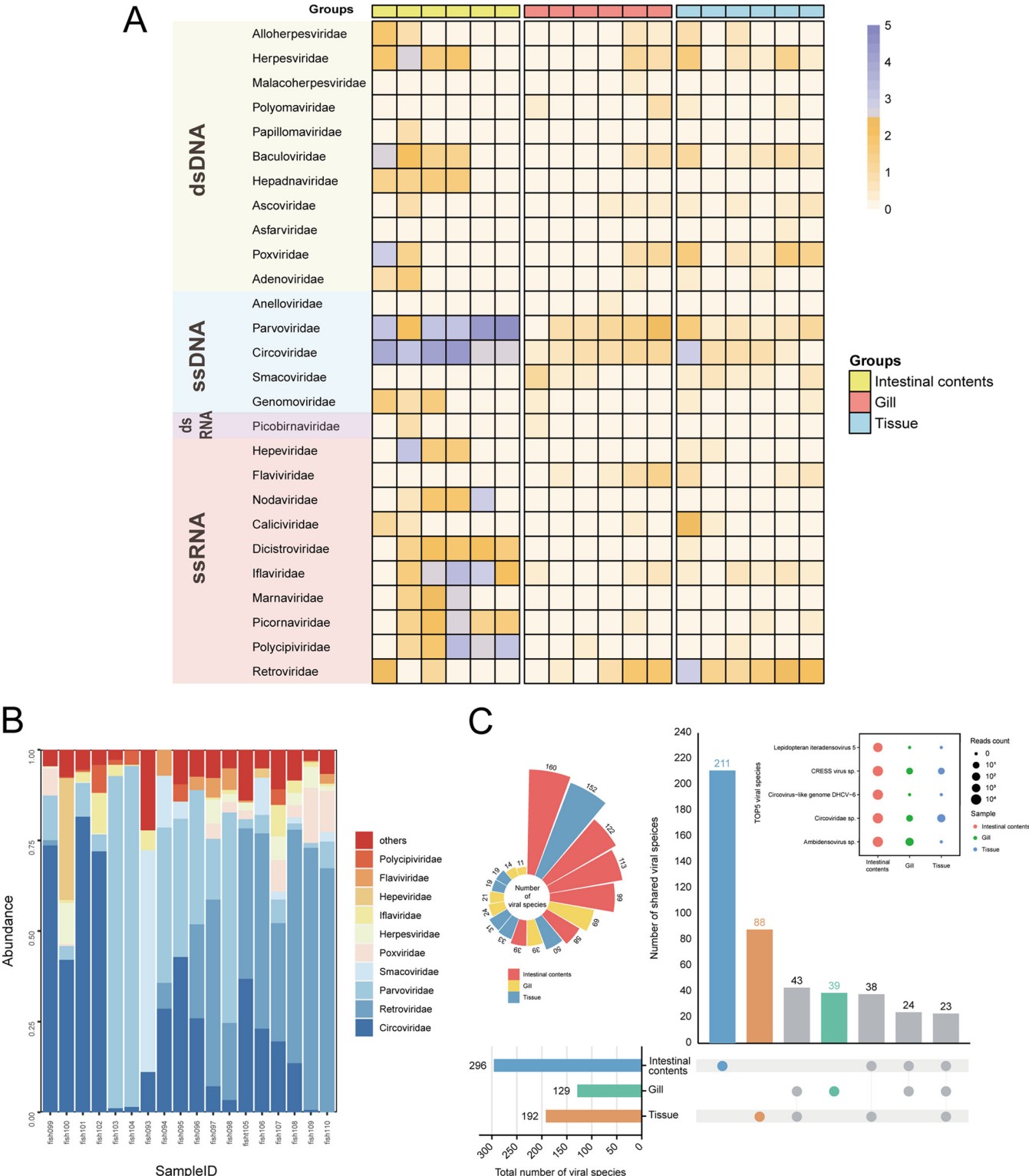

**FIG 2** Taxonomic analyses of viral reads at the level of family or species. (A) Heat map representing the read counts of each viral family of each pool on a $\log_{10}$ scale. Viral types, viral families, and groups are annotated with corresponding colors (see color legend). (B) Bar graphs showing the relative proportion and taxonomy based on viral families. (C) UpSet plot depicting the numbers of shared viral species among the three viromes. Filled spots with interconnecting vertical lines represent sharing between the corresponding libraries. The bars on the left show the total number of viral species in each grouped virome set, while the bars above show the numbers of species within the intersections. A bubble chart shows the top 5 most abundant viral species. Bubble size indicates the abundance of reads assigned to each species. A Nightingale rose diagram illustrates the number of species in each of the 18 pools (see color legend).

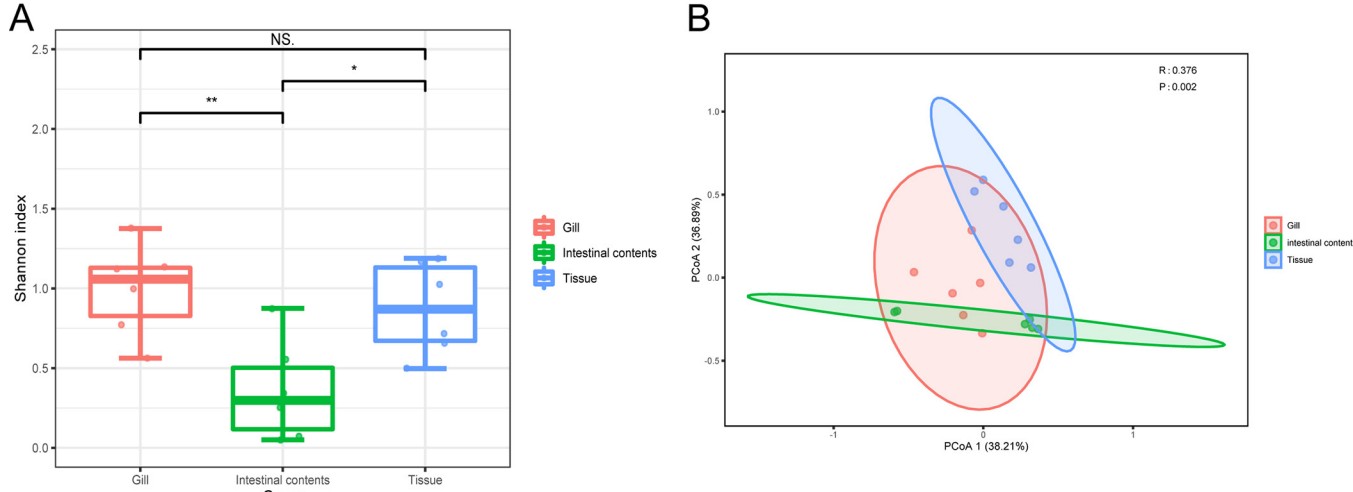

**FIG 3** Diversity of the viral communities between intestinal contents, gills, and tissues. (A) Comparison of virus $\alpha$ diversity was normalized by MEGAN and measured with Shannon index based on viral abundance for intestinal contents, gills, and tissues at the family level. The $P$ value was calculated using the Wilcoxon test. The horizontal bars within boxes represent medians. The tops and bottoms of boxes represent the 75th and 25th percentiles, respectively. *, $P < 0.05$, intestinal contents versus tissue; **, $P < 0.01$, intestinal contents versus gill. (B) Comparison of virus $\beta$ diversity was normalized by MEGAN and PCoA of intestinal contents, gills, and tissues at the family levels. A $P$ value of <0.05 was considered statistically significant.

shown in Fig. 3A. More specifically, compared to the "gill" group and the "tissue" group, the "intestinal contents" group had significantly the lowest $\alpha$ diversity. According to unweighted UniFrac analysis, principal-coordinate analysis (PCoA) revealed that the composition of the viral communities exhibits statistically significant differences across the three groups at the family level based on $\beta$ diversity ($P = 0.002$) (Fig. 3B).

**Novel fish papillomavirus.** Papillomaviruses are nonenveloped, circular, and double-stranded DNA viruses of the *Papillomaviridae* family, which comprises two subfamilies (*Firstpapillomavirinae* and *Secondpapillomavirinae*) (22). The subfamily *Firstpapillomavirinae* infects birds, mammals, reptiles, and other terrestrial vertebrates, and some of these viruses are carcinogenic and cause diseases that are fatal to their hosts (23–25). Since the papillomavirus was discovered in fish for the first time, documented 15 papillomaviruses have also been discovered in fish from seawater or farms in Antarctica, Hungary, and the Mediterranean region, and some individuals have shown multiple papillomas on the skin, albeit this has not yet been found in China (10, 26–28).

We identified novel fish papillomavirus for the first time in the pool Fish100 and verified it in pools Fish94 and Fish106, with a positivity rate of 25.4% (15/59). Notably, one of the fish had papillomavirus found in its all samples (gills, intestinal contents, and tissues). We obtained the whole genome by nested PCR, totaling 6,010 bp, consistent with the length of fish-associated papillomaviruses. It encoded four open reading frames (ORFs) that contained the core and predicted protein products for two early genes, E1 and E2, that encoded protein products involved in replication, two late genes, L1 and L2, that encoded viral capsid proteins, and a protein termed oncoid, which is structurally similar to the human papillomavirus (HPV) E6 or E7 proteins (10). blastx comparison of the GenBank nr database revealed the best match between *Papillomaviridae* fishchang100 (GenBank no. OP933686) and haddock-associated papillomavirus (GenBank no. MZ570862), with amino acid sequence identities of 51.15% and 59.49% in E1 and L1, respectively. For terrestrial vertebrate papillomaviruses, the E1 gene is the most conserved, and this is also true for fish (10). Analysis of papillomavirus E1 proteins from all vertebrate host groups revealed that all fish papillomaviruses are derived from a common ancestor and are highly distinct from other papillomaviruses (Fig. 4; see Fig. S1 in the supplemental material).

**Novel fish hepadnavirus and hepevirus.** The class of viruses known as hepatitis B viruses (HBVs), which are members of the *Hepadnaviridae* family and have reverse-transcribed DNA genomes, are a major human disease with a significant risk of cirrhosis and

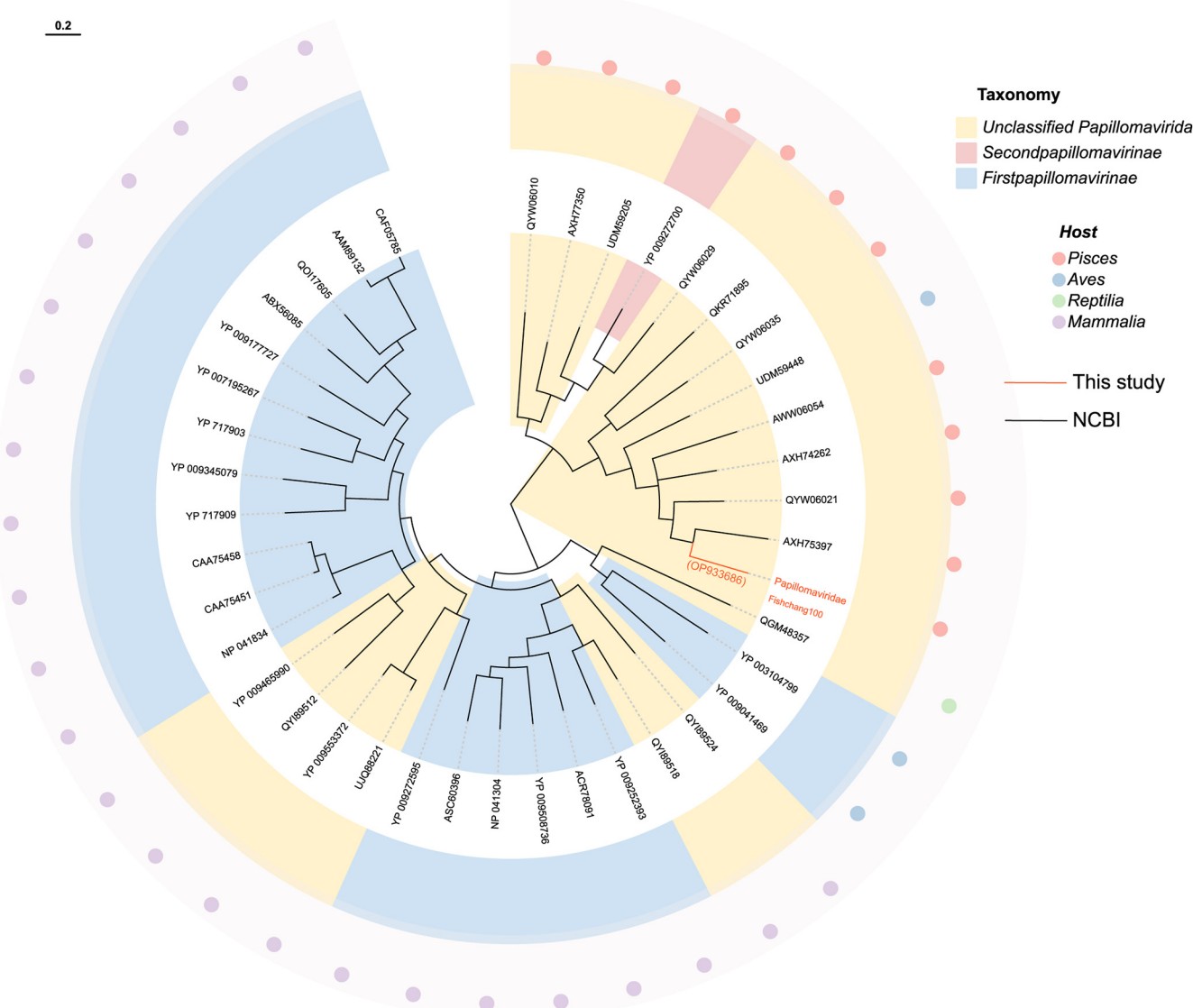

**FIG 4** Phylogenetic relationship of *Papillomaviridae*. The phylogenetic tree is based on E1 protein. The name in orange indicates the sequence obtained in this study. See the legend for relevant labeling.

hepatocellular cancer (29). It is interesting to note that the pool Fish101 included a potentially new hepadnavirus. The more divergent Fish101-hepa-1 (GenBank no. OP933689) shared 50.45% amino acid similarity to white sucker hepatitis B virus (WSHBV) (GenBank no. MW161145). The recently described WSHBV from white sucker (*Catostomus commersonii*) and Fish101-hepa-1 from this study were part of the diverging lineage (termed parahepadnaviruses) in the viral clade (Fig. 5A). These viruses belong to clades that are part of the newly discovered and highly divergent *Nackednavirus*, which have been found in multiple fish species.

In addition to hepatitis B, hepatitis E is an important human disease in several regions, and hepevirus has been detected in fish, birds, and mammals (14). We identified a novel hepevirus in the pool Fish100, which shared 39.66% amino acid similarity to Wenling *Thamnaconus striatus* hepevirus (WTSHV) (GenBank no. MG600006), a sequence found in large batches of fish viruses in 2018 (4). The novel hepevirus sequence Fish100-hepe-1 (GenBank no. OP933684) has 41.63% amino acid identity with the RNA-dependent RNA polymerase (RdRp) gene of WTSHV, tentatively named Lassa fish hepatitis virus (LFHV), which belongs to a new species of the family *Hepeviridae*. The current phylogenetic tree

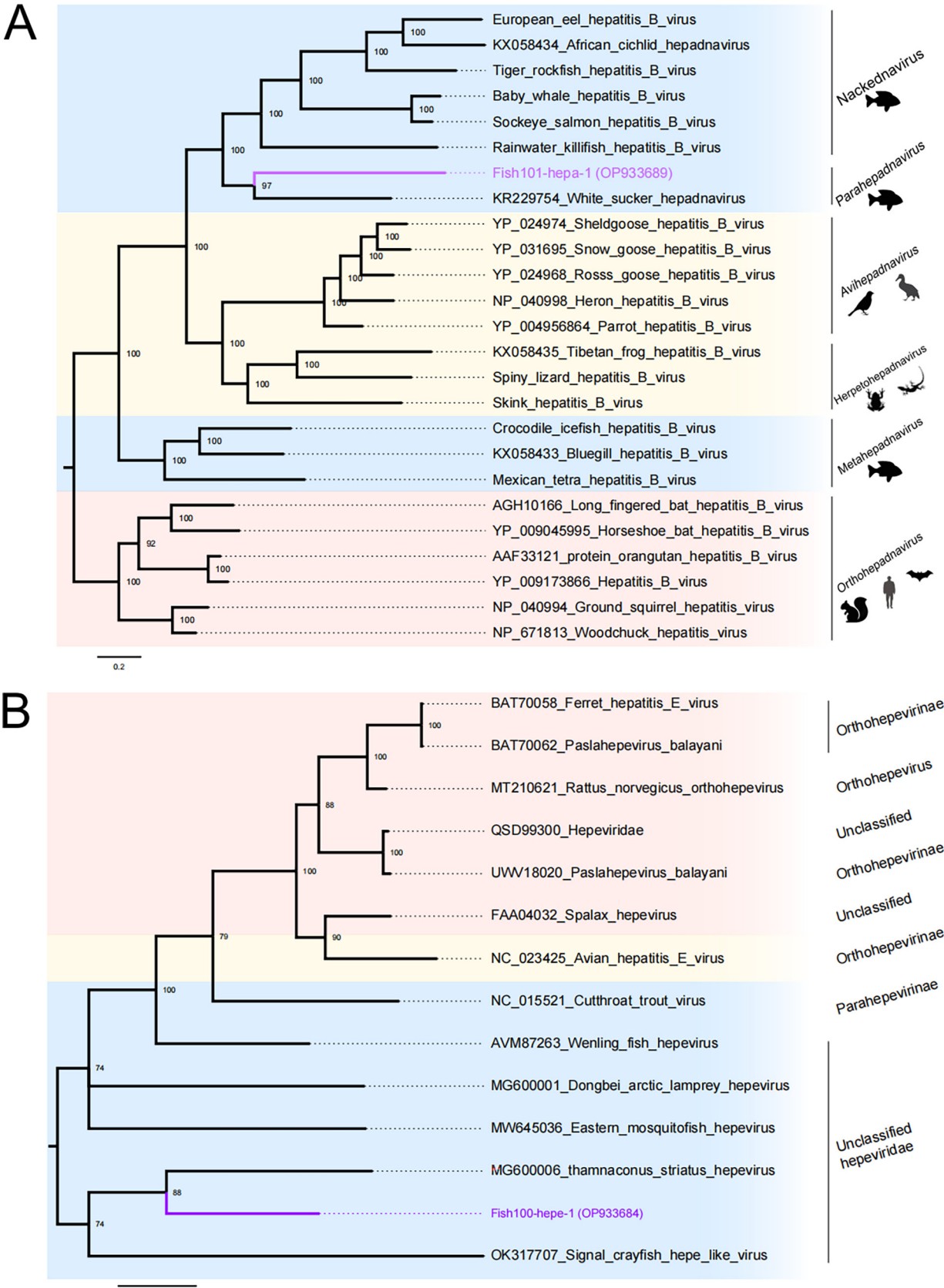

**FIG 5** Phylogenetic relationship of *Hepadnaviridae* and *Hepeviridae*. Host groups are indicated in different colors: mammals in red, birds and reptiles in yellow, and fish in blue. The name of the virus family or genus is shown above each phylogeny, and the name in purple indicates

indicates that LFHV clusters with fish hepeviruses and are classified as "unclassified hepeviridae" (Fig. 5B).

**Fish *Parvoviridae* sequences.** The 4- to 6-kb single-stranded DNA genome of parvoviruses is nonenveloped and typically only has two major ORFs. Three subfamilies make up the *Parvoviridae* family: the invertebrate-infecting *Densovirinae*, the vertebrate-infecting *Parvovirinae*, and the human-infecting *Hamaparvovirinae* (30, 31). However, only in 2019 was the first case of fish parvovirus identified, and it caused a high-mortality outbreak in Hubei tilapia, followed by successive identification of fish parvoviruses in a gulf pipefish (*Syngnathus scovelli*) and zander (*Sander lucioperca*) (32–35), and these pioneering studies suggest that fish may be an as yet undiscovered but very important host for parvoviruses.

With sequence lengths ranging from 2,612 bp to 5,478 bp, and some containing a third major ORF in addition to the two major ORFs encoding nonstructural and capsid proteins, we recovered and characterized a large number of parvoviruses in all pools, with seven ORFs intact and one with only the nonstructural (NS1) protein ORF (Fig. 6A). The classification criteria for viruses in the family *Parvoviridae* are based on NS1 protein coverage of >80% and amino acid sequence identity of >85%, and hence the compared parvoviruses can be regarded as belonging to the same species (31). Additionally, the amino acid sequences of every parvovirus-encoded NS1 protein in a genus are >30% identical to one another. Based on the completed NS1 protein amino acid sequences of representative viruses in the family *Parvoviridae*, a phylogenetic tree was created using these guidelines (Fig. 6B). These sequences were highly divergent and clustered in six clades: five of them named FT105parV1, FI103parV1, FI099V1, FI099V2, and FI099parV4 clustered with viruses recovered from samples collected from bird anal swabs, suggesting that the viruses are likely to have come from fish, which is part of the bird's diet. Unlike the other viruses discovered in this study, one of them, FT105parV1, is more intimately associated with mammals. Remarkably, in the phylogenetic tree of NS1, FI099parV2 was grouped as belonging to the *Densovirinae* subfamily, which infects invertebrates but formed a distinct branch, sharing <50% amino acid identity and <80% coverage with the most closely related virus, proving to be a possible new genus. It is also worth mentioning that since the NS1 protein of FI103parV2 described here bears less than 30% similarity to any currently known parvovirus NS1 protein, it may represent a new genus and a new species of the *Densovirinae* subfamily. In addition to these genomes, which are classified as belonging to the family *Parvoviridae*, the sequences named FI099parV4 and FI099parV1 showed close relationships with known novel parvo-like hybrids––probably of diatom origin––sharing 93.09% and 75.74% amino acid sequence identities, respectively.

**Circular Rep-encoding single-stranded DNA viruses in fish.** The eight fish-derived CRESS DNA (circular Rep-encoding ssDNA) virus strains described in this study can be divided into the following taxonomic groups based on their Rep sequences: *Circoviridae* (*n* = 2), CRESSV3 (*n* = 2), CRESSV5 (*n* = 4), CRESSV6 (*n* = 1), and unclassified CRESS-DNA viruses (*n* = 1). All of these display a high level of genetic diversity (Fig. 7A). The Rep sequences of 8 fish-associated CRESS DNA viruses displayed homology to sequences of fish, invertebrates, wild birds and environmental sequences at the amino acid level. In the CRESSV5 group, for instance, the Rep sequences of three fish-associated circoviruses (FI099cirV1, FI099cirV2, and FI101cirV1) were closely similar to those previously discovered in samples from fish and wild birds, sharing 50.34%, 69.51%, and 64.09% amino acid identities, respectively. These sequences also formed two new branches in the phylogenetic tree, signifying the new genus. In particular, FI099cirV3 was 99.06% identical at the amino acid level of the Rep gene to bat circovirus (GenBank no. KJ641738) collected in Tibet in 2013, suggesting this fish circovirus has a recent common ancestor with a bat circovirus. In addition, FI099cirV3 was detected in the corresponding gills, intestinal contents, and tissues.

**FIG 5** Legend (Continued)
the sequence obtained in this study. Each scale bar indicates the number of amino acid substitutions per site. (A) Bayesian inference tree based on amino acid sequences of polymerase protein of viruses belonging to the family *Hepadnaviridae*; (B) Bayesian inference tree based on amino acid sequences of RdRp of viruses belonging to the family *Hepeviridae*.

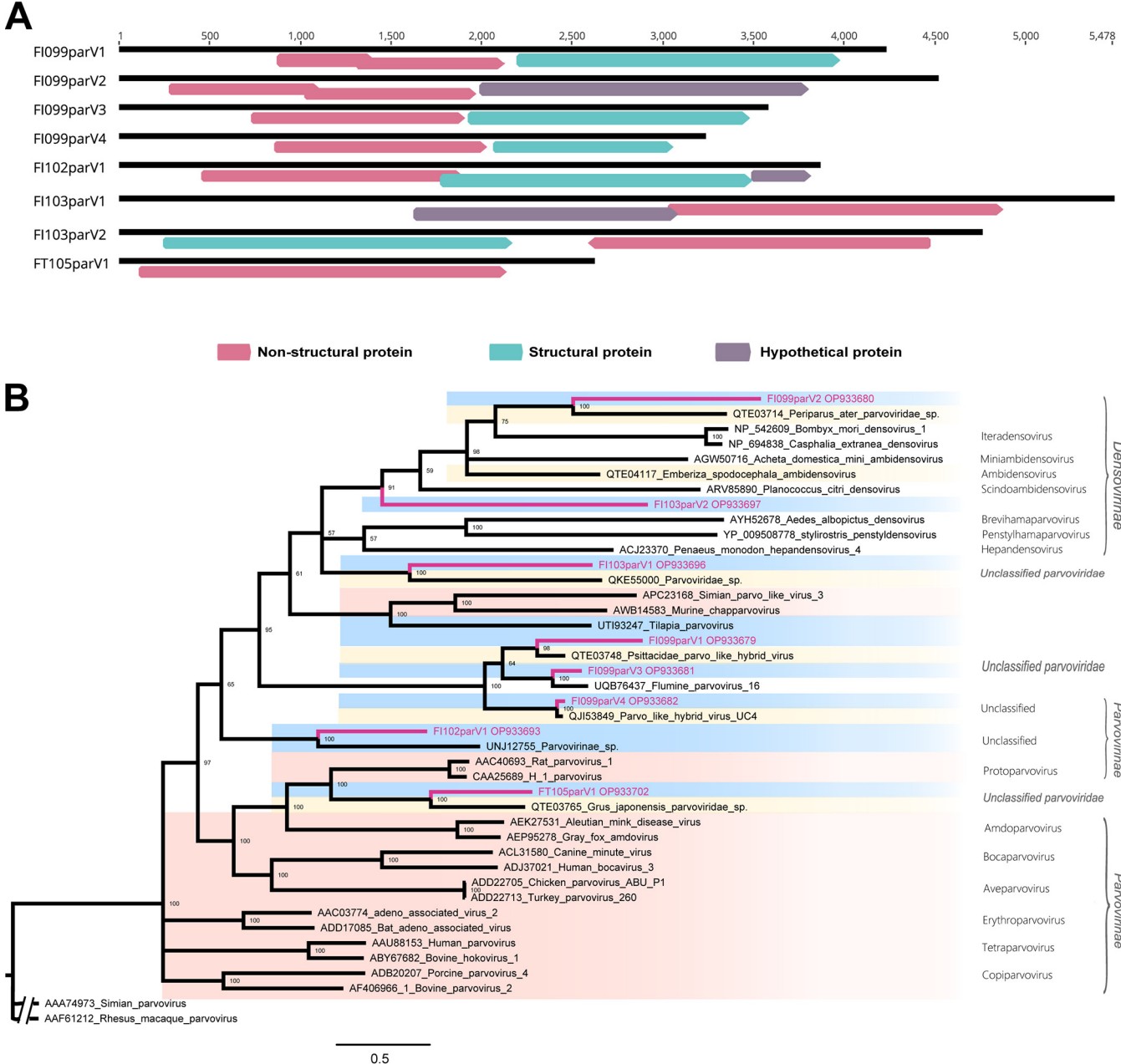

**FIG 6** Phylogenetic relationship of *Parvoviridae*. (A) The genome organization of each virus is shown. (B) Phylogenetic tree based on NS protein. The names in red indicate sequences obtained in this study. Host groups are indicated with different background colors: mammals in red, birds and reptiles in yellow, fish in blue, and invertebrates or environmental samples in white. Each scale bar indicates the number of amino acid substitutions per site.

These circoviruses have a genome size of 1,679 to 2,413 nucleotides (nt) and classical genome organization encoding Rep and Cap proteins in the same direction, whereas circoviruses classified in the CRESSV6 group and unclassified CRESS DNA group have a genome size of 2,832 to 2,922 nt and Rep and Cap proteins in opposite directions (Fig. 7B). Furthermore, blastx analyses revealed that 6 of the 8 fish-associated CRESS DNA viruses identified here shared <70% Rep amino acid sequence identities with their closest relatives.

**Novel RNA viruses in fish.** In the current study, other 8 divergent RNA virus genomes were identified in the samples of fish, which included *Picornaviridae* (*n* = 1), *Iflaviridae* (*n* =1), *Caliciviridae* (*n* = 1), and picorna-like viruses (*n* = 5). Except for the calicivirus, the RdRp amino acid sequence identities among the 8 RNA viruses and their closest relatives in blastx search shared <65% identity to their best matches.

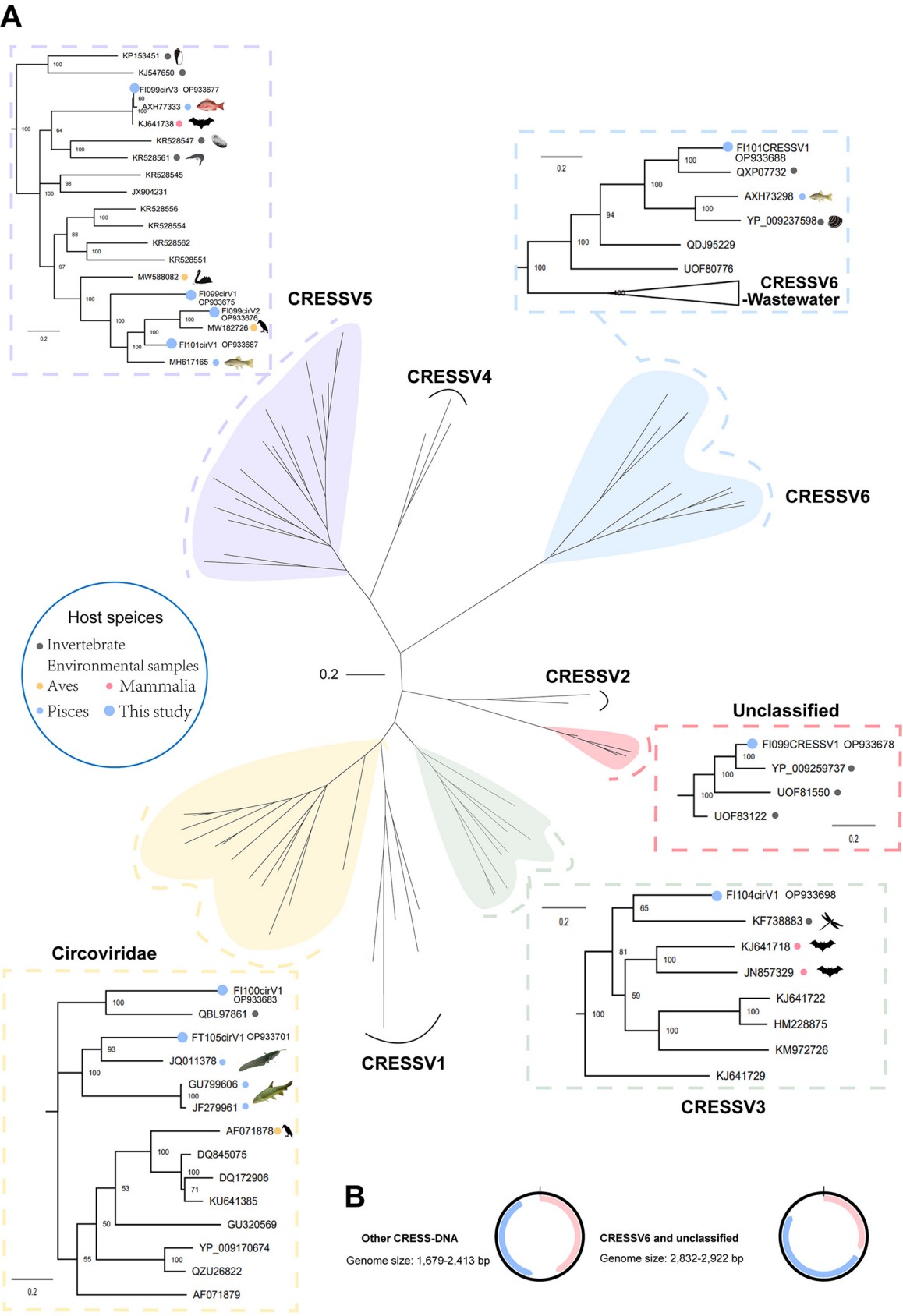

**FIG 7** Phylogenetic relationship of CRESS DNA viruses. (A) Phylogenetic tree based on the Rep amino acid sequences. The viruses identified from fish in this study and the other different host species are marked with dots of corresponding colors (see the color legend). Each scale bar indicates the number of amino acid substitutions per site. (B) Genome organization of CRESS DNA viruses identified in this study.

A novel picornavirus sequence was detected in fish that was related to sequences of other fish picornaviruses (Fig. 8). The novel fish picornavirus FI102picV1 specifically shared 51.72% amino acid identity with the Chinese wenling rattails picornavirus in its RdRp amino acid sequence. Although several picornaviruses, including the ones listed here, have been associated with fish host death, little is known about their epidemiology and disease potential. Caliciviruses are rapidly evolving viruses that have led to outbreaks associated with significant global morbidity and mortality, and there have been cases of massive mortality from caliciviruses in yellow catfish (15). According to the phylogenetic tree, the calicivirus is a brand-new member of a taxon that has previously been found in marine fishes. The family *Iflaviridae* is thought to infect marine fish and was first described in Belize's bonefish (36). The new iflavirus clustered with the invertebrate soybean thrips, according to the phylogenetic tree, nevertheless the RdRp gene shared only a 38.91% similarity with it. A similar event occurred with picorna-like viruses, which mostly clustered in invertebrates and shared 32.11% to 62.46% sequence identity with the best matches.

## DISCUSSION

Recent metagenomic surveys have uncovered an unprecedented diversity of viruses in fish (4). Adequate knowledge of virus diversity in fish is of great importance for viral evolution and enrichment of the fish virus pool, but diversity is still largely unknown for viruses in seemingly healthy fish in the Chinese highlands, which show accelerated rates of evolution at the whole-genome level (37). To help ascertain the presence and evolution of viruses in fish, we found viral genomes from a metagenomic analysis of fish collected from the LR, Tibet, China. In total, our study found 28 potentially novel viral species, 23 of which were distinctly vertebrate associated.

CRESS DNA viruses and parvoviruses were the most widespread among the fish species studied here, and almost all shared weak sequence similarity to known viruses. Interestingly, both CRESS DNA viruses and parvoviruses are single-stranded DNA viruses and nonenveloped, and it is conceivable that these phenotypic characteristics help them survive in harsh plateau aquatic environments. Although parvoviruses have only recently been found in fish, they are known to cause lethal outbreaks of disease (32). The parvoviruses identified here further confirm that fish are potentially important hosts for them and that the high diversity of parvoviruses in fish needs to be further explored. The notable observation was the presence of fish-associated circovirus FI099cir3, which shared the highest amino acid identity (99.06%) in Rep with that of the bat circovirus (GenBank no. KJ641738) and 97.49% identity with that of the red snapper circovirus (GenBank no. AXH77333), indicating the possibility of CRESS DNA viral transmission between fish and mammals.

Since the papillomavirus was originally identified in fish, other papillomaviruses have been identified in fish from seawater or farms, although none have been identified in China as of yet (10). Here, we report a papillomavirus derived directly from intestinal contents and identified as positive in tissues and gills by nested PCR. This finding suggests that fish papillomaviruses are exceedingly diversified and may exist in various forms in additional fish species that have not yet been studied. It also verifies the occurrence of papillomaviruses in Chinese highland fish with lesions that are not readily visible to the naked eye. According to the present phylogenetic tree, all fish papillomaviruses have a common ancestor and have quite different evolutionary relationships and genome sizes from mammalian-infecting papillomaviruses. Further studies on the evolutionary process of papillomaviruses based on biological evolutionary relationships are needed in the future.

Viruses that can cause hepatitis and widespread outbreaks include viruses of the family *Hepadnaviridae* and *Hepeviridae*, particularly the hepatitis E virus, which can also cause foodborne transmission as the number of people eating raw fish increases. Consistent with the literature, this research found that the novel hepadnavirus and hepevirus are also clustered with those previously found in fish, strongly suggesting

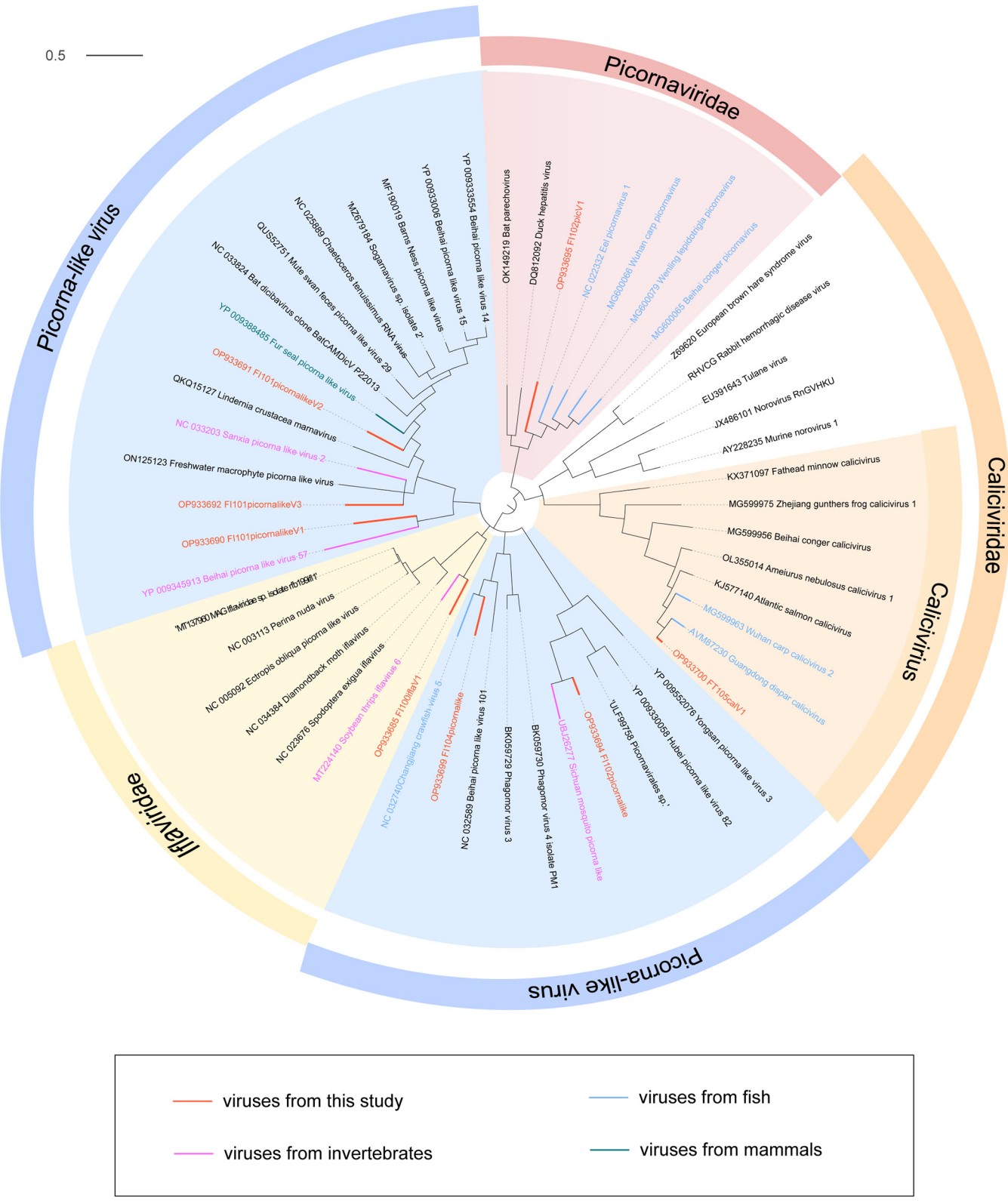

**FIG 8** Phylogenetic relationship of highly divergent RNA viruses. The phylogenetic tree was created based on RdRp amino acid sequences of the RNA viruses. Host groups are indicated with different colors (see color legend).

that fish hepadnavirus and hepevirus may be far more widespread than already sampled. Of particular note, the viruses newly described in fish showed similar tissue tropisms that were comparable to those of their mammalian counterparts, which again argues for their antiquity. For example, among the viruses discovered here, that of the *Hepadnaviridae* family was mainly found in the liver.

Large-scale analyses of viral diversity in vertebrate phylogeny have also shown a high diversity of viruses in fish, usually belonging mainly to viruses in birds and mammals, suggesting an ancient virus-host association (4). Specifically, this study shows that fish RNA viruses are diverse and include many new viruses, not the least of which are those that cause fish virus disease outbreaks, such as calicivirus and picornavirus. Regarding their epidemiology and potential for disease, including those mentioned below, little is known. Additionally, viruses like the iflavirus and picorna-like viruses, which are more frequently linked to invertebrate illnesses, have been discovered in fish. In a collection of tissue samples, viruses belonging to the *Iflaviridae* family, including FI100iflaV1, have been found. In the phylogeny, the novel iflavirus is most closely related to an iflavirus identified in insects. Iflaviruses may be infected with internal parasites, thus indirectly reflecting the parasite load, which seems to be the case in mice (38). Whether this is the case for infection in fish, on the other hand, requires further study.

Although this preliminary study focused only on fish in highland rivers, we identified a total of 28 potentially unknown viruses of nine species overall and found most to be potentially new viruses. Since the fish used for the analysis were taken immediately from the river, we also investigated viruses with a brief incubation period or very low abundance within the host. This supports the idea that fish contain a variety of viruses that will eventually come to light with more thorough sampling. The limited geographical distribution of the fish that were sampled, however, poses a drawback to our investigation. We anticipate that the examination of fish from the more diverse rivers of the highlands will reveal a greater viral variety in fish and allow for comparisons with viruses from non-highland fish. Our study highlights the remarkable diversity of fish virus communities and enriches the current fish virus database. However, further studies are needed to reveal the role of fish viruses in potentially threatening animal- and human-pathogenic mechanisms.

## MATERIALS AND METHODS

**Fish sample collection.** A total of 32 fish were randomly collected in 2021 from LR, the biggest tributary of the YTR, which rises in China's Tibet Autonomous Region. There was no visible exterior pathology on the fish. To broaden viral diversity, intestinal contents, gills, and tissues (liver, muscle, swim bladder, heart, brain) of fish were immediately dissected and stored separately in a −80°C freezer. Some tissue samples were sampled secondarily. The samples are collected and sent to the laboratory immediately. All 292 samples were grouped into 18 pools by the classification of gills, intestinal contents, and tissues (Table 1). The fish species were identified by amplification and Sanger sequencing of the fragment of the control region D-loop. They are classified as the family Cyprinidae (see Table S3 in the supplemental material). All 292 samples from each pool were homogenized, three times frozen, and thawed on dry ice, and then 100 mg of each resuspended in 1 mL of Dulbecco's phosphate-buffered saline (DPBS). The centrifugation (10 min, 15,000 × *g*, 4°C) was followed by the collection of the supernatants. The sample collection and all experiments in this study received ethical approval from Jiangsu University's ethics committee.

**Sample preparation and library construction.** To get rid of large cell-sized particles, 500 $\mu$L of each supernatant was passed through a 0.45-$\mu$m-pore syringe filter (Millipore) and centrifuged at 13,000 × *g* for 5 min. This step collected 166.5 $\mu$L of filtrate enriched in viral particles. The filtrate was treated at 37°C for 60 min with DNase and RNase enzymes (Turbo DNase from Thermo Fisher Scientific, USA; BaselineZeroDNase from Epicentre, USA; Benzonase nuclease from Novagen, USA; and RNase A from Thermo Fisher Scientific) to break down unprotected nucleic acid (39–41). Under the direction of the manufacturer, total nucleic acids (total RNA and DNA) shielded from nuclease digestion within viral capsids were extracted using the QiAamp viral RNA minikit. On the viral nucleic acid samples, reverse transcription procedures, including reverse transcriptase (Super-Script IV; Invitrogen) and random hexamer primers, were carried out. A single cycle of DNA synthesis using Klenow fragment polymerase ensued (New England BioLabs). For ssDNA viruses, ssDNA was converted to dsDNA using the Klenow reaction, and all of the dsDNA products were used to construct libraries. Then, 18 pools were made using the Illumina Nextera XT DNA sample preparation kit, each of which included dual barcoding for different samples. These pools were then sequenced on the Illumina NovaSeq 6000 platform with 250-bp paired-end reads (42).

**TABLE 1** Sampling information of fish in this study

| Pool name | No. of samples | Included sample(s) |
|---|---|---|
| Fish093 | 7 | Gill |
| Fish094 | 7 | Gill |
| Fish095 | 7 | Gill |
| Fish096 | 7 | Gill |
| Fish097 | 7 | Gill |
| Fish098 | 7 | Gill |
| Fish099 | 6 | Intestinal contents |
| Fish100 | 6 | Intestinal contents |
| Fish101 | 7 | Intestinal contents |
| Fish102 | 6 | Intestinal contents |
| Fish103 | 7 | Intestinal contents |
| Fish104 | 7 | Intestinal contents |
| Fish105 | 44 | Liver, muscle, swim bladder, heart, brain |
| Fish106 | 46 | Liver, muscle, swim bladder, heart, brain |
| Fish107 | 40 | Liver, muscle, swim bladder, heart, brain |
| Fish108 | 35 | Liver, muscle, swim bladder, heart, brain |
| Fish109 | 26 | Liver, muscle, swim bladder, heart, brain |
| Fish110 | 23 | Liver, muscle, swim bladder, heart, brain |

**Bioinformatics analysis.** The 250-bp paired-end reads generated for each pool were debarcoded using vendor software from Illumina. Clonal reads were eliminated, and tails with low sequencing quality were trimmed using the Phred quality score of 30 (Q30) as the threshold. The cleaned reads were then compared to an internal nonvirus nonredundant protein database using a DIAMOND blastx search with default parameters to weed out false-positive viral hits (43). The taxonomic classification of the DIAMOND results was then parsed using MEGAN, and the lowest common ancestor (LCA) assignment technique was then done using the default settings. Geneious Prime v.2019.0 (Biomatters, Ltd.) was used to *de novo* assemble every viral sequence read (44). To identify the virus kinds and exclude fake virus sequences, the contigs and singlet sequences were then compared against the viral proteome database using blastx (E value of $<10^{-5}$). The virus blastx database was compiled using the NCBI virus reference proteome ([ftp://ftp.ncbi.nih.gov/refseq/release/viral/](ftp://ftp.ncbi.nih.gov/refseq/release/viral/)) and viral protein sequences from NCBI nr FASTA files (based on annotation taxonomy in the virus kingdom) (45, 46). To avoid false-positive viral hits, candidate viral sequences are compared to an in-house nonvirus nonredundant (NVNR) protein database, which was compiled from nonviral proteins in the NCBI nr fasta file (eliminating sequences from the virus kingdom). The blastx similarity of contigs without significant hits is then compared to viral protein families in the vFam database using HMMER3 to identify more remote viral protein similarities (47, 48). Geneious Prime software and the findings of the blastx search were combined to estimate ORFs in the viral genome. The NCBI Conserved Domain Search was used to find and annotate the protein domains (E value of $<10^{-5}$) (49).

**Viral sequence acquisition and PCR validation.** In order to produce intriguing viral genomes or segments, *de novo* assembly and reference mapping in Geneious Prime v.2019.0 were performed on the assembled contigs and unassembled reads in known taxonomy assignments obtained from the preceding stage. Geneious Prime v.2019.0 was also used for genome annotation, ORF prediction, and primer creation. PCR confirmation was performed for papillomavirus, hepevirus, and hepadnavirus in different samples. Inverse PCR was used to generate the complete genome of the novel papillomavirus. Sequences and characteristics of the primers used in the present study are shown in Table S2. The Sanger method was used for sequencing the PCR products. pMD18-T vectors (TaKaRa Bio, Inc.) were used to clone purified positive PCR results, which were then transformed into DH5$\alpha$ competent cells. Agarose plates containing the transformed DH5$\alpha$ cells were coated with the antibiotic ampicillin, and the plates were incubated at 37°C for 14 h. Bacterial liquid PCR and sequencing were used to identify the positive clones. By quickly amplifying cDNA ends with the SMARTer random amplification of cDNA ends (RACE) 5′/3′ kit (TaKaRa Bio, Inc.) and using the purified PCR result to create expression vectors with In-Fusion, the 3′-terminal region of the fish picornavirus was identified (TaKaRa Bio, Inc.).

**Statistical analysis.** The statistical analysis was carried out using R v.4.2.1 and MEGAN v.6.22.2. Utilizing MEGAN, the composition analysis results from 18 pools were standardized and contrasted (50). The R v.4.2.1 packages pheatmap and vegan were used to depict the viral community structure and richness findings, and the R v.4.2.1 package ggplot2 was used to display the difference in viral communities. If the *P* value was $<0.05$, it was regarded as statistically significant.

**Phylogenetic analysis.** Phylogenetic analyses were performed based on the predicted protein sequences of viruses identified in this study and protein sequences of reference strains belonging to different groups of viruses that were downloaded from the NCBI GenBank database. Related protein sequences were aligned using MUSCLE in MEGA v.10.1.8 with the default setting (51). Bayesian inference trees were then constructed using MrBayes v.3.2.7 (52). We used two concurrent runs of Markov chain Monte Carlo (MCMC) sampling in MrBayes and set "prset aamodelpr=mixed" for the phylogenetic analysis based on protein sequences. The runs were terminated when the split frequency standard deviation was less than 0.01, and the first 25% of the trees were burned in. Additionally, maximum likelihood trees

were built to support all of the Bayesian inference trees in the MEGA software (51). FigTree v.1.4.4 (http://tree.bio.ed.ac.uk/software/figtree/), Adobe Illustrator 2022 v.26.0.1, and iTOL v.6 (53) were used to display the phylogenetic trees.

**Quality control.** To rule out the possibility of nucleic acid contamination in the lab, sterile double-distilled water (ddH$_2$O) (Sangon Biotech) was prepared and processed further under the same conditions as a blank control. Common precautions were followed throughout the process to prevent cross-contamination and nucleic acid degradation. DNase and RNase were absent from every substance that came into direct contact with nucleic acid samples. RNase inhibitors and diethyl pyrocarbonate (DEPC)-treated water (Sangon Biotech) were used to dissolve the nucleic acid samples.

**Data availability.** GenBank has received all genome sequences determined in this study and stored them under accession no. OP933675 to OP933702. The Sequence Read Archive (SRA) has received quality-filtered sequence reads that are listed under the BioProject ID no. PRJNA905657 and BioSample ID no. SAMN31880338.

## SUPPLEMENTAL MATERIAL

Supplemental material is available online only.

**SUPPLEMENTAL FILE 1**, PDF file, 0.5 MB.

## ACKNOWLEDGMENTS

This research was supported by National Key Research and Development Programs of China (grant no. 2022YFC2603801) and the Independent Project of Chengdu Research of Giant Panda Breeding (grant no. 2020CPB-C11).

W.Z., T.S., and P.S. designed the study and methods. X.X., H.Z., K.Q., and M.Z. constructed the libraries. Y.X., X.J., X.W., Q.S., S.Y., L.J., Y.L., and W.Z. completed the data analysis. The paper's first draft was prepared by Y.X. and substantially reviewed and revised by all authors.

We declare no conflict of interest.

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
