## [Reviewer comments · Microbiology Spectrum]

Microbiology Spectrum

Viromics reveals the high diversity of viruses from fishes on the Tibet highland

Yuan Xi, Xiaojie Jiang, Xinrui Xie, Min Zhao, Han Zhang, Kailin Qin, Xiaochun Wang, Yuwei Liu, Shixing Yang, Quan Shen, Likai Ji, Peng Shang, Wen Zhang, and Tongling Shan

Corresponding Author(s): Wen Zhang, Jiangsu University

Review Timeline:

Submission Date:	March 6, 2023
Editorial Decision:	March 27, 2023
Revision Received:	April 23, 2023
Accepted:	May 9, 2023

Editor: Biao He

Reviewer(s): Disclosure of reviewer identity is with reference to reviewer comments included in decision letter(s). The following individuals involved in review of your submission have agreed to reveal their identity: Kanchan Bhardwaj (Reviewer #1); Santanu Chattopadhyay (Reviewer #2)

Transaction Report:

DOI: <https://doi.org/10.1128/spectrum.00946-23>

March 27, 2023

Dr. Wen Zhang
Jiangsu University
School of Medicine
301 Xuefu Road, Zhenjiang, Jiangsu 212013, PR China;
Zhenjiang, Jiangsu 212013
China

Re: Spectrum00946-23 (Vriomics reveals the high diversity of viruses from fishes on the Tibet highland)

Dear Dr. Wen Zhang:

Thank you for submitting your manuscript to Microbiology Spectrum. Your manuscript has been read by two experts. In light of their comments, I would like to invite you to revise the manuscript. When submitting the revised version of your paper, please provide (1) point-by-point responses to the issues raised by the reviewers as file type "Response to Reviewers," not in your cover letter, and (2) a PDF file that indicates the changes from the original submission (by highlighting or underlining the changes) as file type "Marked Up Manuscript - For Review Only". Please use this link to submit your revised manuscript - we strongly recommend that you submit your paper within the next 60 days or reach out to me. Detailed instructions on submitting your revised paper are below.

Link Not Available

Sincerely,

Biao He

Journals Department
Reviewer comments:

Reviewer #1 (Public repository details (Required)):

Data has been submitted to GenBank and is accessible.

Reviewer #1 (Comments for the Author):

In this study, authors have performed metagenome analysis of viruses that were collected from 32 apparently healthy fish found in the Yarlung Tsangpo River. Viruses were extracted from 3 types of samples i.e the intestinal contents, gills and a sample containing tissues from multiple organs (liver, muscle, swim bladder, heart, brain). The analyses included: (i) taxonomic annotation of the recovered viral sequences; (ii) determination of alpha diversity in each sample type; (iii) determination of beta

viral diversity among intestinal contents, gills and the sample containing tissues from multiple organs; and (iv) phylogenetic analysis of select viruses.

Authors report:

1. Presence of over 400 different virus species of the families Circoviridae, CRESS-DNA viruses, Hepadnaviridae, Parvoviridae, Papillomaviridae, Hepeviridae, Caliciviridae, Picornaviridae, Iflaviridae and unclassified picorna-like viruses, in all the analyzed samples. In the intestinal contents and gills, majority of the reads belong to the Circoviridae and Parvoviridae families. Whereas, in the sample with tissues from multiple organs, majority of the reads belong to Retroviridae family.
2. Identification of a novel papillomavirus, a hepadnavirus and a hepevirus.

I have following comments:

1. Most of the manuscript should be re-written with clear sentences and better connectivity between different sections. For instance, "Results" section includes identification of novel papillomavirus, hepadnavirus and hepevirus but only papillomavirus is mentioned in the abstract.
2. Ln 1, Spelling "Vriomics" to be corrected.
3. Ln 110-115, What is the purpose of this analysis? In the following section (Ln 118), there is mention of 27 viral families.
4. Ln 122, how is viral abundance determined?
5. Ln 147, what is the interpretation of the results in figure 3B?
6. Ln 332, "Sample Preparation" would be an appropriate title than "Viral Metagenomic Analysis". Ln 333-343, More details about sample preparation for sequencing should be provided because it is surprising to me that viral nucleic acid could be extracted in sufficient quantity from 500 ul filtrate. Also, what was done to sequence ssDNA genomes?
7. Ln 352, "the contigs and singlet sequences were then compared against the viral proteome database". Please specify the viral proteome database used here.
8. The section "Viral sequences acquisition and PCR validation" Ln 357-367, needs to be re-written to explain what is being done and why. Why was PCR validation done only for papillomavirus and not for hepadnavirus and hepevirus. Similarly, the section "Phylogenetic analysis" Ln 376-378, need to be re-written with clarity.
9. "CRESS-DNA viruses" does not need to be italicized.
10. Ln 120, "Circoviridae" Ln 351, Ln 357 "de novo" to be italicized.

Reviewer #2 (Public repository details (Required)):

The authors have mentioned :

"GenBank has received all genome sequences and stored them there under the accession numbers OP933675-OP933702. The Sequence Read Archive (SRA) has received quality-filtered sequence reads that are listed under the BioProject ID PRJNA905657 and the BioSample ID SAMN31880338".

Reviewer #2 (Comments for the Author):

My overall opinion is that the manuscript is attempting an important issue by virome analysis, particularly if we keep in mind the possibility of zoonosis. Any virome analysis is challenging and the challenges include higher cost as well as lack of standardized universal techniques. In the present study, the authors followed strategies of pulling the samples in 18 groups to reduce the cost and went for direct metagenomic sequencing without purification of the viruses from the fish parts. I appreciate the strategy, but the manuscript has several major issues that need to be addressed.

The specific points:

1. The importance of the study could have been increased if the authors could tell us which species of fishes are carrying which RNA viruses. The study tells us which parts of the fishes have the possibilities of carrying which RNA viruses, but it does not tell the name of the fish since they have pulled the samples. However, I hope that the authors have kept the samples and can go back to the original samples and can identify the species with at least RT-PCR. Without this, the study will be a weak study. For example, would I not want to know which fish is carrying the papillomavirus? I would. But the manuscript tells only about "pool-fish 100" and the list in the Table1 shows it is "intestinal content". But which fish? The authors need to make a table showing which viruses have been isolated from which fish and from which body part.
2. Although good a sequencing method-Illumina NovaSeq 6000 and 250-bp paired-end reads- have been used, I am not totally sure if the depth was good enough for virome analysis particularly when no viral enrichment was done. In line 103 it was mentioned that "In most of the 18 pools, the observed virus species flatten out, it can be assumed that the sequencing depth has covered all species in the sample, and increasing the sequencing data can no longer increase diversity". But, in Fig 1A one of the lines (possibly from the fish tissues) are still going up in a straight line. One of the lines has a sharp bend. Few lines had a sudden stop. Authors need to address these things.

3. Line 120. "The Cirocoviridae family had a significantly higher number of viral reads than any other viral families (66,818 reads, accounted for 53.18%)". However, in line 101 the authors have mentioned that they have identified 197,306 viral reads. If there were 197,306 viral reads, the 66818 reads does not account for 53.18%. It would be 33.89%.

4. Fig 3B. Do the circles show the correct overlap? I think that they do not show the correct overlap. Particularly for the tissue, which is shown in blue.

5. Line 159. "We identified novel fish papillomavirus for the first time in pool-fish100 and verified it in gills, intestinal contents, and tissues with a positive rate of 25.4% (15/59), with one fish positive for its all samples to determine its infection". Very interesting since the "pool-fish100", according to Table 1, comes from the gut. The question comes, why the authors have not found papillomavirus in the pool-fish 93-98 and 105-110?

6. Line 170. "groups revealed that all fish papillomaviruses are derived from a common ancestor and are highly distinct from other papillomaviruses". The distinctiveness of the fish papillomaviruses is not very clear from the circular tree. The authors are requested to present an uprooted tree also. This will clarify if the fish papillomaviruses are truly distinct or have some similarities with other branches.

7. Line 328. "The samples from each pool were homogenized, three times frozen and thawed on dry ice, and then resuspended in 1 mL of Dulbecco's Phosphate Buffered Saline (DPBS). How much tissue was there in 1 ml?

8. Line 35. "we conducted a metagenomic survey of healthy fishes and sampled intestinal contents, gills, and tissues". This information is important for the abstract, not for the importance.

9. DNA viruses were not studied.

10. The manuscript is not carefully written. For example, the title "Vriomics reveals the high diversity of viruses from fishes 1 on the Tibet highland" is wrong. It should be Viromics, not Vriomics. Several other mistakes are there in the manuscript. If the manuscript gets accepted for publication it should go through a thorough scrutiny.

Staff Comments:

Preparing Revision Guidelines

Please return the manuscript within 60 days; if you cannot complete the modification within this time period, please contact me. If you do not wish to modify the manuscript and prefer to submit it to another journal, please notify me of your decision immediately so that the manuscript may be formally withdrawn from consideration by Microbiology Spectrum.

In this study, authors have performed metagenome analysis of viruses that were collected from 32 apparently healthy fish found in the Yarlung Tsangpo River. Viruses were extracted from 3 types of samples *i.e* the intestinal contents, gills and a sample containing tissues from multiple organs (liver, muscle, swim bladder, heart, brain). The analyses included: (i) taxonomic annotation of the recovered viral sequences; (ii) determination of alpha diversity in each sample type; (iii) determination of beta viral diversity among intestinal contents, gills and the sample containing tissues from multiple organs; and (iv) phylogenetic analysis of select viruses.

Authors report:

1. Presence of over 400 different virus species of the families *Circoviridae*, CRESS-DNA viruses, *Hepadnaviridae*, *Parvoviridae*, *Papillomaviridae*, *Hepeviridae*, *Caliciviridae*, *Picornaviridae*, *Iflaviridae* and unclassified picorna-like viruses, in all the analyzed samples. In the intestinal contents and gills, majority of the reads belong to the *Circoviridae* and *Parvoviridae* families. Whereas, in the sample with tissues from multiple organs, majority of the reads belong to *Retroviridae* family.
2. Identification of a novel papillomavirus, a hepadnavirus and a hepevirus.

I have following comments:

1. Most of the manuscript should be re-written with clear sentences and better connectivity between different sections. For instance, "Results" section includes identification of novel papillomavirus, hepadnavirus and hepevirus but only papillomavirus is mentioned in the abstract.
2. Ln 1, Spelling "Vriomics" to be corrected.
3. Ln 110-115, What is the purpose of this analysis? In the following section (Ln 118), there is mention of 27 viral families.
4. Ln 122, how is viral abundance determined?
5. Ln 147, what is the interpretation of the results in figure 3B?
6. Ln 332, "Sample Preparation" would be an appropriate title than "Viral Metagenomic Analysis". Ln 333-343, More details about sample preparation for sequencing should be provided because it is surprising to me that viral nucleic acid could be extracted in sufficient quantity from 500 ul filtrate. Also, what was done to sequence ssDNA genomes?
7. Ln 352, "the contigs and singlet sequences were then compared against the viral proteome database". Please specify the viral proteome database used here.
8. The section "Viral sequences acquisition and PCR validation" Ln 357-367, needs to be re-written to explain what is being done and why. Why was PCR validation done only for papillomavirus and not for hepadnavirus and hepevirus. Similarly, the section "Phylogenetic analysis" Ln 376-378, need to be re-written with clarity.
9. "*CRESS-DNA viruses*" does not need to be italicized.
10. Ln 120, "*Circoviridae*" Ln 351, Ln 357 "de novo" to be italicized.

Point-by-point responses to the issues raised by the reviewers:

(The comments of the reviewers are in italics which are followed by our responses.)

Reviewer comments:

Reviewer #1 (Comments for the Author):

In this study, authors have performed metagenome analysis of viruses that were collected from 32 apparently healthy fish found in the Yarlung Tsangpo River. Viruses were extracted from 3 types of samples i.e the intestinal contents, gills and a sample containing tissues from multiple organs (liver, muscle, swim bladder, heart, brain). The analyses included: (i) taxonomic annotation of the recovered viral sequences; (ii) determination of alpha diversity in each sample type; (iii) determination of beta viral diversity among intestinal contents, gills and the sample containing tissues from multiple organs; and (iv) phylogenetic analysis of select viruses.

Authors report:

1. Presence of over 400 different virus species of the families Circoviridae, CRESS-DNA viruses, Hepadnaviridae, Parvoviridae, Papillomaviridae, Hepeviridae, Caliciviridae, Picornaviridae, Iflaviridae and unclassified picorna-like viruses, in all the analyzed samples. In the intestinal contents and gills, majority of the reads belong to the Circoviridae and Parvoviridae families. Whereas, in the sample with tissues from multiple organs, majority of the reads belong to Retroviridae family.

2. Identification of a novel papillomavirus, a hepadnavirus and a hepevirus.

=Response: Thank you for your nice comments on our article. According to your suggestions, we have supplemented several data here and corrected several mistakes in our previous draft. The detailed point-by-point responses are listed below.

Comments:

1. Most of the manuscript should be re-written with clear sentences and better connectivity between different sections. For instance, "Results" section includes identification of novel papillomavirus, hepadnavirus and hepevirus but only papillomavirus is mentioned in the abstract.

=Response: Thank you for your suggestion. We have revised most of the sentences that had unclear expressions, and marked the changes in red. (For example, please see the revised edition: Ln 26-28, Ln 46-48).

2. Ln 1, Spelling "Vriomics" to be corrected.

=Response: We feel sorry for our carelessness. We have corrected it and we also feel great thanks for your point out (Please see the revised edition: Ln 1).

3. Ln 110-115, What is the purpose of this analysis? In the following section (Ln 118), there is

mention of 27 viral families.

=Response: To be more clearly, we have added a more detailed interpretation regarding this analysis (Please see the revised edition: Ln 113-114 and Ln 121). This analysis (Ln 113-118) aims to elaborate a total of 27 specific and nearly complete viral sequences obtained by *de novo* assembly and reference mapping, while Ln 121 describes the number of viral families in all pools by classifying viral reads for which no further analysis has been performed.

4. Ln 122, how is viral abundance determined?

=Response: Thank you for your question. The number of viral reads in different pools was referred to as viral abundance. The legend and title of Fig. 2A state that the "intestinal contents" group has the greatest viral abundance and the "gill" group, the least (Please see the revised edition: Ln 125-126).

5. Ln 147, what is the interpretation of the results in figure 3B?

=Response: Thank you for your question. According to the data in figure 3B, based on beta diversity, there are statistically significant differences in the viral communities' composition between the "intestinal contents", "gill", and "tissue" groups at the family level ($P=0.002$) (Please see the revised edition: Ln 150-152).

6. Ln 332, "Sample Preparation" would be an appropriate title than "Viral Metagenomic Analysis". Ln 333-343, More details about sample preparation for sequencing should be provided because it is surprising to me that viral nucleic acid could be extracted in sufficient quantity from 500 ul filtrate. Also, what was done to sequence ssDNA genomes?

=Response: Thank you for your suggestion. We have replaced the title with "Sample preparation and library construction" (Please see the revised edition: Ln 336). Our previous research has reported numerous cases and verified this method (1–3), and we had added some details in this section (Please see the revised edition: Ln 337-339, Ln349). For ssDNA viruses, ssDNA was converted to dsDNA using the Klenow reaction and all the dsDNA products were used to construct libraries (Please see the revised edition: Ln 346-347).

7. Ln 352, "the contigs and singlet sequences were then compared against the viral proteome database". Please specify the viral proteome database used here.

=Response: Thank you for your suggestion. We have added specific viral proteome database

(<ftp://ftp.ncbi.nih.gov/refseq/release/viral/>) and other databases (Please see the revised edition: Ln 359-365).

8. The section "Viral sequences acquisition and PCR validation" Ln 357-367, needs to be re-written to explain what is being done and why. Why was PCR validation done only for papillomavirus and not for hepadnavirus and hepevirus. Similarly, the section "Phylogenetic analysis" Ln 376-378, need to be re-written with clarity.

=**Response:** Thank you for your suggestion. According to your comments, we have re-written the section "Viral sequences acquisition and PCR validation" and added supplemental information: Table S2 (Please see the revised edition: Ln 371-376 and Table S2). We also have re-written the section "Phylogenetic analysis" (Please see the revised edition: Ln 388-391).

9. "CRESS-DNA viruses" does not need to be italicized.

=**Response:** Thank you for your suggestion. We have corrected it and we also feel great thanks for your point out (Please see the revised edition: Ln 115).

10. Ln 120, "Circoviridae" Ln 351, Ln 357 "de novo" to be italicized.

=**Response:** Thank you for bringing it to our attention; we have modified it (Please see the revised edition: Ln 123, Ln 114, Ln 357 and Ln369).

Reviewer #2 (Public repository details (Required)):

The authors have mentioned :

"GenBank has received all genome sequences and stored them there under the accession numbers OP933675-OP933702. The Sequence Read Archive (SRA) has received quality-filtered sequence reads that are listed under the BioProject ID PRJNA905657 and the BioSample ID SAMN31880338".

Reviewer #2 (Comments for the Author):

My overall opinion is that the manuscript is attempting an important issue by virome analysis, particularly if we keep in mind the possibility of zoonosis. Any virome analysis is challenging and the challenges include higher cost as well as lack of standardized universal techniques. In the present study, the authors followed strategies of pulling the samples in 18 groups to reduce the

cost and went for direct metagenomic sequencing without purification of the viruses from the fish parts. I appreciate the strategy, but the manuscript has several major issues that need to be addressed.

=Response: Thank you for your nice comments on our article. According to your suggestions, we have supplemented several data here and corrected several mistakes in our previous draft. The detailed point-by-point responses are listed below.

The specific points:

1. The importance of the study could have been increased if the authors could tell us which species of fishes are carrying which RNA viruses. The study tells us which parts of the fishes have the possibilities of carrying which RNA viruses, but it does not tell the name of the fish since they have pulled the samples. However, I hope that the authors have kept the samples and can go back to the original samples and can identify the species with at least RT-PCR. Without this, the study will be a weak study. For example, would I not want to know which fish is carrying the papillomavirus? I would. But the manuscript tells only about "pool-fish 100" and the list in the Table1 shows it is "intestinal content". But which fish? The authors need to make a table showing which viruses have been isolated from which fish and from which body part.

=Response: Thank you for your suggestion. We have added the taxonomy of fish. The fish species were identified by amplification and Sanger sequencing of the fragment of control region D-loop. They are classified as the family Cyprinidae (Please see the revised edition: Line 329-331). We also added the isolation of main viruses in Table S3.

2. Although good a sequencing method-Illumina NovaSeq 6000 and 250-bp paired-end reads have been used, I am not totally sure if the depth was good enough for virome analysis particularly when no viral enrichment was done. In line 103 it was mentioned that "In most of the 18 pools, the observed virus species flatten out, it can be assumed that the sequencing depth has covered all species in the sample, and increasing the sequencing data can no longer increase diversity". But, in Fig 1A one of the lines (possibly from the fish tissues) are still going up in a straight line. One of the lines has a sharp bend. Few lines had a sudden stop. Authors need to address these things.

=Response: Thank you for your suggestion. We apologized for the limitations in sample coverage and sequencing. The sequencing depth of few pools were insufficient but most of the libraries are sufficient to capture almost all known viral species in the samples.

3. Line 120. "The Cirocoviridae family had a significantly higher number of viral reads than any other viral families (66,818 reads, accounted for 53.18%)". However, in line 101 the authors have mentioned that they have identified 197,306 viral reads. If there were 197,306 viral reads, the

66818 reads does not account for 53.18%. It would be 33.89%.

=Response: We feel sorry for our carelessness. We have corrected it and we also feel great thanks for your point out (Please see the revised edition: Line 124).

4. Fig 3B. Do the circles show the correct overlap? I think that they do not show the correct overlap. Particularly for the tissue, which is shown in blue.

=Response: Thank you for your suggestion. We have modified and double-checked by R studio and we also feel great thanks for your point out (Please see the revised edition: Fig 3).

5. Line 159. "We identified novel fish papillomavirus for the first time in pool-fish100 and verified it in gills, intestinal contents, and tissues with a positive rate of 25.4% (15/59), with one fish positive for its all samples to determine its infection". Very interesting since the "pool-fish100", according to Table 1, comes from the gut. The question comes, why the authors have not found papillomavirus in the pool-fish 93-98 and 105-110?

=Response: Thank you for your suggestion. To make it clearer, we replaced this sentence with "We identified novel fish papillomavirus for the first time in pool-fish100 and verified it in pool-fish94 and -fish106 with a positive rate of 25.4% (15/59). Notably, one of the fish had papillomavirus found in its all samples (gills, intestinal contents, and tissues)." (Please see the revised edition: Line 162-164).

6. Line 170. "groups revealed that all fish papillomaviruses are derived from a common ancestor and are highly distinct from other papillomaviruses". The distinctiveness of the fish papillomaviruses is not very clear from the circular tree. The authors are requested to present an uprooted tree also. This will clarify if the fish papillomaviruses are truly distinct or have some similarities with other branches.

=Response: Thank you for your suggestion. We agree with you and have presented an uprooted tree in Fig.S1 (Please see the revised edition: Fig S1 and Line 174).

7. Line 328. "The samples from each pool were homogenized, three times frozen and thawed on dry ice, and then resuspended in 1 mL of Dulbecco's Phosphate Buffered Saline (DPBS). How much tissue was there in 1 ml?"

=Response: Thank you for your question. 100 mg tissue was resuspended in 1 mL DPBS individually (Please see the revised edition: Line 331-332).

8. Line 35. "we conducted a metagenomic survey of healthy fishes and sampled intestinal contents, gills, and tissues". This information is important for the abstract, not for the importance.

=Response: Thank you for your suggestion. We agree with you and have deleted it in the section "Importance" and modified the section "Abstract" (Please see the revised edition: Line 22-23 and Line 34-37).

9. DNA viruses were not studied.

=Response: Thank you for your suggestion but we have described the DNA viruses in several

parts: “Novel fish papillomavirus” (Line 153-174), “Novel fish hepadnavirus” (Line 175-183), “Fish parvoviridae sequences” (Line 193-221), “CRESS DNA viruses” (Line 222-240).

10. The manuscript is not carefully written. For example, the title "Vriomics reveals the high diversity of viruses from fishes 1 on the Tibet highland" is wrong. It should be Viromics, not Vriomics. Several other mistakes are there in the manuscript. If the manuscript gets accepted for publication it should go through a thorough scrutiny.

=Response: Thank you for your suggestion. We apologized for our careless, we have scrutinized the whole article and marked some modifications as red (For example, please see the revised edition: Line 1, Line 123, Line 227).

References:

1. Zhang W, Li L, Deng X, Kapusinszky B, Pesavento PA, Delwart E. 2014. Faecal virome of cats in an animal shelter. *Journal of General Virology* 95:2553–2564.
2. Zhang W, Li L, Deng X, Blümel J, Nübling CM, Hunfeld A, Baylis SA, Delwart E. 2016. Viral nucleic acids in human plasma pools. *Transfusion* 56:2248–2255.
3. Zhang W, Yang S, Shan T, Hou R, Liu Z, Li W, Guo L, Wang Y, Chen P, Wang X, Feng F, Wang H, Chen C, Shen Q, Zhou C, Hua X, Cui L, Deng X, Zhang Z, Qi D, Delwart E. 2017. Virome comparisons in wild-diseased and healthy captive giant pandas. *Microbiome* 5:90.

May 9, 2023

Dr. Wen Zhang
Jiangsu University
School of Medicine
301 Xuefu Road, Zhenjiang, Jiangsu 212013, PR China;
Zhenjiang, Jiangsu 212013
China

Re: Spectrum00946-23R1 (Viromics reveals the high diversity of viruses from fishes on the Tibet highland)

Dear Dr. Wen Zhang:

I am pleased to inform you that your manuscript has been accepted, and I am forwarding it to the ASM Journals Department for publication. You will be notified when your proofs are ready to be viewed.

Sincerely,

Biao He
Editor, Microbiology Spectrum

Journals Department
Authors have satisfactorily addressed the points raised by me.